# Novel *spa* and Multi-Locus Sequence Types (MLST) of Staphylococcus Aureus Samples Isolated from Clinical Specimens in Korean

**DOI:** 10.3390/antibiotics8040202

**Published:** 2019-10-29

**Authors:** Yae Sung Mun, You Jin Hwang

**Affiliations:** 1Department of Life science, College of Bio-nano technology, Gachon University, 191, Hambangmoe-ro, Yeonsu-gu, Incheon 21936, Korea; yae01@naver.com; 2Department of Health Sciences and Technology, GAIHST, Gachon University, Incheon 21999, Korea; 3Department of Biomedical Engineering, College of Health Science, Gachon University, 191, Hambangmoe-ro, Yeonsu-gu, Incheon 21936, Korea

**Keywords:** *spa* typing, *spa*, MLST, multidrug-resistance, methicillin-resistant *Staphylococcus aureus* (MRSA)

## Abstract

*Staphylococcus aureus* bacteremia is one of the most frequent and severe bacterial infections worldwide. The increased incidence of *S. aureus* infections with a diverse pattern of *S. aureus* protein A (*spa*) types across different geographic regions is a global challenge. This study investigated a novel *spa* type of methicillin-resistant *S. aureus* in a clinically isolated specimen. A total of 109 clinical *S. aureus* samples were subjected to 19 sets of antimicrobial susceptibility testing using the Kirby–Bauer disk diffusion method. Molecular typing was performed with *S. aureus* protein A (*spa*) and multi-locus sequence types (MLST) via polymerase chain reaction and sequencing. The methicillin-resistant *S. aureus* samples in our study accounted for 55.05% (60/109) of the total. A novel *spa* type was detected in five (5/60) strains. This *gh2*2 isolate was identified in antimicrobial susceptibility tests of 15 kinds of antibiotics. Antibiotic resistance genes included *mecA,*
*TEM*, *aac(6′)-aph(2”)*, *ermA,* and *tetM.* Eleven *S. aureus* samples were classified as *t2460*, *t338*, *t324*, *t693*, five unknown *spa* types (new *spa* types), and undefined MLST (novel MLST). We report a high prevalence rate of *t2460* methicillin-resistant *S. aureus* samples in our country. Additionally, novel *spa*
*gh2*2, MLST ST4613, and clonal compact CC5-type strains (T1:M1:B1:B1:M1:E1:K1, r26:r17:r34:r34:r17:r13:r16, mlst;1:4:1:4:559:495:10) showing multidrug resistance were identified among *S. aureus* samples.

## 1. Introduction

Methicillin-resistant *Staphylococcus aureus* (MRSA) is a prominent pathogen that causes severe infections in both healthcare and community settings [1]. MRSA is associated with considerable infection worldwide [2]. Resistance to antimicrobial agents, especially nosocomial pathogens, has become one of the most serious challenges worldwide. Excessive therapeutic application of antimicrobial agents in both humans and animals has contributed to the development of widespread antibiotic resistance in bacteria [3,4]. Multidrug-resistant *S. aureus* is a serious public health concern warranting medical attention [5].

Epidemiological studies are an essential component in the study of clonality, evolutionary pathways, genetic diversity of pathogens, and the spread of *S. aureus* infections [6]. Various molecular typing methods can be used to type MRSA isolates [7]. Spa typing is based on the number of tandem repeats and the sequence variation in region X of the protein A gene. The *spa* gene contains three distinct regions: Fc, X, and C [8,9]. Based on a literature review, the *spa* type distribution of MRSA strains isolated from patients in different geographic locations worldwide exhibits diverse patterns [10]. Pulsed-field gel electrophoresis (PFGE) with high discriminatory power is the documented gold standard among the various DNA sequencing methods available [11,12]. However, due to its labor-intensive methodology, difficulties associated with data exchange between laboratories, and the need for inter-laboratory standardization, PFGE has been replaced by multi-locus sequence typing (MLST) and staphylococcal protein A (*spa*) typing [8]. Of these, *spa* typing, which relies only on the assessment of the number and sequence variation in repeats at the X region of the *spa* gene, exhibits excellent discriminatory power and has become a user-friendly and cost-effective typing tool with a standardized nomenclature [5,12,13,14].

MLST is an effective tool for evolutionary investigation and differentiation of isolates according to nucleotide variations in seven housekeeping genes [15]. MLST has proven very useful for macroepidemiology and evolutionary studies [16]. However, *spa* typing is also both easier and less costly to perform than MLST or PFGE. Therefore, *spa* typing combined with unambiguous typing results of MLST can reduce costs for laboratories over time.

Country-specific data relating to the distribution of diverse *spa* and MLST types have yet to be clearly reported. This study elucidated the patterns of antibiotic resistance based on antibiotic sensitivity testing and the molecular epidemiology and characterization, including new *spa* and novel MLST typing approaches in *S. aureus* samples collected from clinical sources at Gil Hospital, Incheon, Korea.

## 2. Results

We tested for antimicrobial susceptibility using Kirby–Bauer disc diffusion and established the isolates as resistant or susceptible to antimicrobial agents based on the diameters of the inhibition zone. Our susceptibility testing showed that 56.88% (60/109) of *S. aureus* samples were resistant to methicillin. We selected 13 specific samples and compared the susceptibility results via disc diffusion. The antibiotic-resistant genes in *S. aureus* samples were identified using the PCR assay (Table 1).

Our susceptibility testing also showed (Table 1) that two *S. aureus* samples (*gh11*, *gh20*) were susceptible to penicillin G, tetracycline, erythromycin, and extended spectrum antibiotics, including cephalosporins, cabapenems, and monobactams. Four samples (*gh2*, *gh22*, *gh19*, *gh21*) exhibited resistance against methicillin, penicillin G, erythromycin, and partially extended spectrum cephalosporins, cabapenems, and monobactams (Table 1).

Three samples (*gh2*, *gh22*, and *gh19*) tested PCR-positive for *TEM*, *mecA*, *SCCmec* type II, *aac(6′)-aph(2”)*, *tet*M, and *erm*A genes (Table 2). Sample *gh21* was tested positive for *bla*_TEM_, *mecA*, *SCCmec* type II, *tet*M, and *erm*A, but not for *aac(6′)-aph(2”)*, *kanamycin*, or *gentamicin* antibiotic genes.

Sequence analysis showed 13 specific samples and revealed different *spa* types among the 11 clinical samples of *S. aureus*, including four known *spa* types (*t2460*(2), *t338*, *t324*, and *t693*) and six new *spa* types (*gh2*, *gh11*, *gh19*, *gh20*, *gh21*, and *gh22*), as defined by the Ridom SpaServer. These results, involving multiple patterns among the distribution of *spa* types, are presented in Figure 1 and Table 2.

Six new *spa* types were identified in Ridom SpaServer. Spa type *t2460* was the predominant *spa* type in our samples (data not shown here), and *spa* sequence analysis showed five different genotypes among 11 specific MRSA isolates (Figure 2). Four *spa* types, namely *t2460*, *t338*, *t324*, and *t693*, were distinguished using the Ridom method in this study (Table 2).

We compared a total of 35 known *spa* strains of MRSA, with 13 of our isolate samples obtained from patient samples in Korea. A phylogeny review of 46 total split circles depicting the phylogeny tree revealed a region containing 35 known isolates and 11 samples (Figure 3) (multiple sequence alignment by CLUSTALW, http://rest.genome.jp/link/. Kyoto University Bioinformatics Center, Kyoto, Japan).

Spa gene samples *gh2* and *gh19* showed similar sequences and a single partially different sequence carrying the same repeat sequence of Ridom types. Sample *gh22* was similar to *gh2* and *gh19*, except for a single variation in a repeat sequence. The repeat numbers assigned by the SpaServer were 26-17-34-34-17-13-16 and 26-17-34-34-17-13-314 (Table 2). Further, *gh2*, *gh19*, and *gh22* were the only six nucleotide sequence variations carrying a single repeat sequence variation and similar superantigen profiles, suggesting that the strains *gh2*, *gh19*, and *gh22* were closely linked. Additionally, *gh11*, *gh20*, and *gh21* represented different repeat sequences. The repeat numbers assigned by the SpaServer were 07-06-17-21-81-34-22-34, 07-23-12-21-298-254, and 26-17-34-34-298-377-298-298-298-314.

MLST analysis showed 13 specific samples. They were differently revealed and undefined MLST types (Table 3). Three strains (*gh2*, *gh21*, *gh22*) exhibited ST4613, as well as clonal complex CC5. Ten strains did not have defined MLST and had different clonal complexes.

The phylogenetic MLST tree showed minimum spanning based on MLST allelic profiles. Nine category samples were neighbor-joining, and each sample was compared with subtype sequences (Figure 3). The *gh7* and *gh11* samples were arcC, aroE and glpF different sequences but had the same type of pubMLST classification. The *gh19* strain showed different similarity distances to *gh7*, *gh11*, *gh2*7, *gh97*, and *gh100* strains, but was more similar than *gh2*, *gh5*, and *gh20* strains (Figure 3). Figure 3 shows the MLST sequence variant distance and antibiotic resistance. The red mark indicates multiple antibiotics resistance and green indicates no or little resistance.

## 3. Discussion

The widespread emergence of MRSA (*S. aureus)* is a great public health challenge. Currently, the spread of MRSA limits therapeutic options and causes severe morbidity and mortality in hospitalized patients [2,3]. The prevalence of MRSA also varies widely in different geographic regions worldwide [3,17,18]. Molecular spa typing of *S. aureus* isolates is important for the identification of dominant strains associated with disease outbreaks and drug resistance, and also in tracing the transmission chain and source of infection [4,19]. In this study, 11 specific MRSA isolates were found that had not been reported by other studies from Korea. According to the results of drug susceptibility testing, the expected MRSA strains were characterized by higher rates of resistance to antimicrobial agents, including penicillin G, tetracycline, erythromycin, extended spectrum antibiotics, cephalosporins, cabapenems, and monobactams. The *gh2*, *gh22* and *gh19* MRSA strains showed high rates of antibiotic resistance and tested positive for antibiotic-resistant genes.

The SpaServer identified 18,514 different *spa* types as of January 28, 2019. Among the 109 MRSA and MSSA strain isolates, we found five novel *spa* types. The *spa* type *t2460* was the most dominant type (20/60, 33.3%) in the studied population. Interestingly, the *t2460* genotype was only found among the MRSA isolates. The isolated strain *t2460* exhibiting *SCCmec* type II was predominant among the investigated MRSA isolates. Similar studies reported individual *t2460* MRSA isolates in human hosts from China and Korea [20,21,22].

In this study, we identified new *spa* types *gh2*, *gh11*, *gh19*, *gh20*, *gh21*, *gh22*, and *SCCmec* type II in our samples. Four *spa* types, namely *t2460*, *t338*, *t324*, and *t693*, were distinguished using the Ridom type method in this study (Table 3). In another international study, the prevalence of *t037*, which was the most frequent *spa* type in 2000–2001, was reduced in 2007–2008. By contrast, *t2460* was the most frequent *spa* type in 2007–2008, as shown in [21]. The increased prevalence of *t2460* strain in 2007–2008 was not related to the outbreak. Similarly, *t002*, *t062*, *t601*, and *t2460* exhibit *spa*-CC002 [21,22]. In Asian countries, the most predominant clones in the region included ST59-MRSA-*SCCmec* type IV-*spa* type *t437* in Taiwan, Hong Kong, Vietnam, and Sri Lanka; ST30-MRSA-*SCCmec* type IV-*spa* type *t019* in the Philippines; and ST72-MRSA-*SCCmec* type IV-*spa* type *t324* in Korea [23]. Some of the health care-associated (HA-MRSA) isolates from Taiwan, the Philippines, and Korea showed the same genotypic characteristics as community-associated MRSA (CA-MRSA) isolates in these countries [6,24,25]. In China, *t030* was the predominant *spa* type [26]. However, it was also reported in Iran as the fifth most common *spa* type. Moreover, *t037* was the second most common *spa* type in Asia, and was reported in several Asian countries (Korea, China, Taiwan, Iran, and Malaysia) [21,22,23,24,25,26,27,28].

Previous MLST results of molecular typing showed only a few sequence types (ST) in Korea. In these results, our samples were the similar types ST4613, ST544, ST188, and ST30. Cha et al. reported on ST239 and the presence of only a few major MRSA strains in the clonal spread of MRSA in Korea [8]. Testing for the prevalence of MLST types showed that ST5-*SCCmec* II-*t2460* and ST5-*SCmec* II-*t002* (68.4%) were the predominant clones, followed by ST72-*SCCmec* IV-*t324* (15.8%), ST239-*SCCmec* III-*t037* (10.5%), and ST1-*SCCmec* IV-*t286* (3.9%) [29].

Both ST59 and ST338 belong to CC59, which have shown the prevalent CA-MRSA clone in China and other Asian countries [30,31]. Along with three CC59 MRSA isolates, we also identified a single ST59 MSSA isolate among the Ready-to-eat (RTE) food samples. In China, ST59 and ST338 were the first and second most dominant sequence types (STs) in cases of pediatric community-acquired pneumonia [32]. In recent years, community-associated MRSA (CA-MRSA) has emerged as an important cause of infection, with geographical differences among strains—ST1 (USA400) and ST8 (USA300) exist in North America, ST80 is found in Europe, ST59 is found in the Asia-Pacific region, and ST30 is noted worldwide [33,34]. Among the MSSA strains, ST188, ST72, ST5, and ST30 occurred most frequently, consistent with previous findings in Korea [17,35,36]. ST72, which is a major CA-MRSA clone in South Korea, was distinct from those that have spread throughout Asia (ST30, ST59, and ST338) or internationally [36]. The Korean CA-MRSA strain has emerged in the community and has recently been spreading in healthcare settings [36,37].

In this study, five new *spa* and ten new MLST types were reported for the first time and did not represent the currently known strains of MRSA, MSSA, or antibiotic susceptibility. In summary, the results showed that *t2460* was one of the major *spa* types along with *SCCmec* type II in Korea. Our results demonstrated differences in the regional *spa* and MLST types of Korea compared with those originating in Asia, Europe, and other geographic locations.

## 4. Materials and Methods

### 4.1. Materials and Bacterial Isolates

Our study was conducted at Gachon University Gil Medical Center in Incheon, South Korea, between April 2016 and June 2018. The research was approved by the ethics committee of Gil Hospital, Gachon University of Medicine. Sample identification and antimicrobial susceptibility testing of *S. aureus* isolated from blood agar plates (Shin Yang Chemical Co., Ltd.’s media., Seoul, Korea) were performed using a MicroScan Pos Breakpoint Combo panel type 28 (PBC28; Beckman Coulter, West Sacramento, CA, USA). Sample strains were streaked onto sheep blood agar (Sinyang Diagnostics, Seoul, Korea) and transported to our laboratory after culture. One colony was picked from each blood agar plate and incubated in lysogeny broth with shaking (80 rpm) at 37 °C overnight. Isolates were preserved in 20% glycerol (*vol/vol*) and stored in a −80 °C freezer until further use.

### 4.2. Antimicrobial Susceptibility Testing

We tested for antimicrobial susceptibility using the Kirby–Bauer disc diffusion method described by Clinical and Laboratory Standard Institute (CLSI) guidelines, 2013 [38]. Each bacterial suspension was adjusted to McFarland 0.5 turbidity, swabbed onto Muller–Hinton agar, and incubated in the presence of antibiotic discs at 35 °C for 18 h.

We tested the following 19 antibiotic discs from Liofilchem (Liofilchem, Roseto degli Aburzzi, Italy): penicillin G (10 IU), methicillin (5 μg), kanamycin (30 μg), gentamicin (10 μg), streptomycin (10 μg), tetracycline (30 μg), erythromycin (15 μg), vancomycin (30 μg), chloramphenicol (30 μg), amoxicillin (25 μg), ticarcillin (75 μg), piperacillin (100 μg), cefepime (30 μg), cefotaxime (30 μg), ceftazidime (30 μg), imipenem (10 μg), ertapenem (10 μg), meropenem (10 μg), and aztreonam (30 μg). We measured the diameters of the inhibition zones and determined each isolate as resistant or susceptible to antimicrobial agents based on CLSI guidelines and Liofilchem quality control parameters. We obtained *S. aureus* control strain *Staphylococcus aureus* ATCC 29,213 (Korean Culture Center of Microorganisms, Seodaemun-gu, Seoul, Korea).

### 4.3. Genomic DNA Extraction

The QIAamp DNA Mini Kit (Cat No: 51306, Qiagen GmbH, Hilden, Germany) was used for genomic DNA extraction according to the manufacturer’s instructions. DNA concentrations were determined using a NanoDrop^TM^ spectrophotometer (Thermo Fisher Scientific, Waltham, MA, USA).

### 4.4. Identification of MecA, Bla_TEM,_ and the Detection of Genes Associated with Antimicrobial Resistance by Multiplex PCR

The PCR primers used to detect *mecA*, *bla*_TEM,_ and antimicrobial resistance genes are listed in references [39,40,41]. The following reaction mixture was added to each sample: 10 pmol of each primer, 2 μL DNA (100 ng), and 10 μL iQ^TM^ cyber (SYBR)^®^ Green supermix (2× reaction buffer with dNTP(dATP, dCTP,dTTP, dGTP)s, iTaq DNA polymerase, SYBR^®^ Green I, fluorescein, and stabilizers; Bio-Rad, Hercules, CA, USA). The volume was adjusted to 20 μL with autoclaved, triple-distilled water. The PCR conditions in the thermal cycler (TC-512, UK) were as follows: 94 °C for 3 min, followed by 35 cycles of denaturation at 94 °C for 30 s, annealing at 56 °C for 30 s, and extension at 72 °C for 45 s. The reaction was ended with a final extension step at 72 °C for 10 min. PCR products were subjected to electrophoresis using 2% agarose gel in 1× TBE buffer at 100 V for 25 min. The 100 bp DNA ladder (Bioneer, Daejeon, Korea) was used as a molecular size marker. PCR products in gels were visualized with a Safe Green loading dye (Applied Biological Materials Inc., Vancouver, BC, Canada).

### 4.5. spa, MLST Typing, and Phylogenetic Analysis

The *spa* typing was performed as described by Harmsen et al. (8). The polymorphic X region of the *spa* gene was amplified using primers *spa*1095F (5′-AGACGATCCTTCGGTGAGC-3′) and *spa*1517R (5′-GCTTTTGCAATGTCATTTACTG-3′). PCR *spa* gene products were subjected to DNA sequencing of both strands by Macrogen (Seoul, Korea). The sequences were analyzed using Ridom StaphType v2.0.3 software (Ridom GmbH). The guidelines derived from the Ridom SpaServer database (http://www.spaserver.ridom.de) were used to assign the edited sequences to the particular *spa* types. The relationships between *spa* types were investigated using the Based Upon Repeat Pattern *(*BURP) clustering algorithm [42] incorporated into Ridom StaphType. Sequences were analyzed using multiple sequence alignment by the CLUSTALW program (https://www.genome.jp/tools-bin/clustalw, Kyoto University Bioinformatics Center).

The primer design for the seven multi-locus sequence typing (MLST) housekeeping genes (*arcC*, *aroE*, *glpF*, *gmk*, *pta*, *tpi*, and *yqiL*) was obtained from the MLST database (http://www.mlst.net/). We used PCR primers composed of 10 pmol of upstream primer and 10 pmol of downstream primer, 100 ng/mL of template, and 10 uL of 2× iQ^TM^ SYBR^®^ Green supermix (Bio-Rad, Hercules, CA, USA). Sterile water was added to achieve a volume of 20 uL. PCR cycling was performed as follows: 95 °C for 5 min; followed by 30 cycles of 94 °C for 30 s; 55 °C for 30 s; 72 °C for 1 min; and a final extension step of 72 °C for 10 min. The products of seven housekeeping gene fragments were sequenced (Bioneer, Daejeon, Korea) and were compared with allele profiles from the *S. aureus* sample.

MLST database (http://www.mlst.net/) and sequence types (STs) were derived and analyzed with eBURST software (http://saureus.mlst.net/eburst/). Isolates that shared six of seven MLST loci were considered as belonging to the same clonal complex (CC).

## Figures and Tables

**Figure 1 antibiotics-08-00202-f001:**
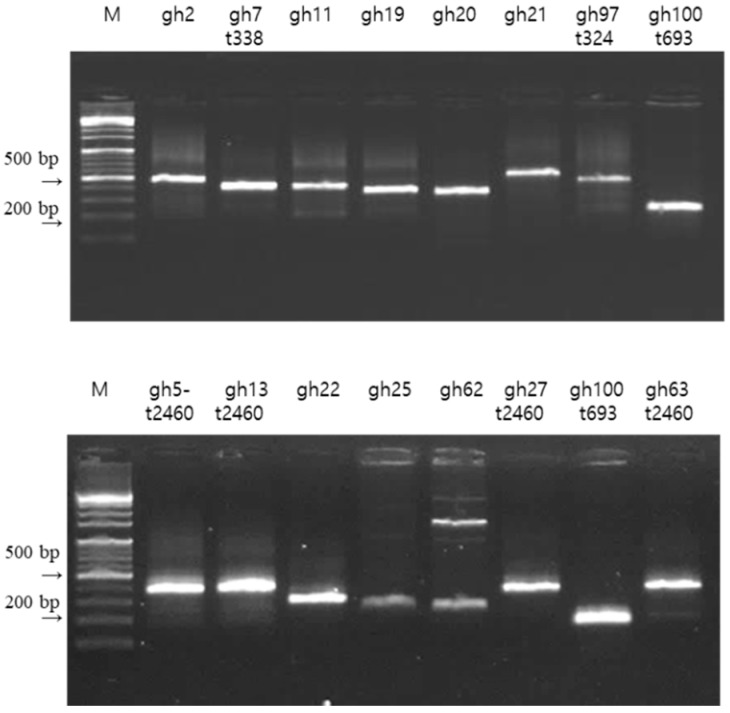
PCR amplification of the *spa* type gene. The PCR results were visualized by agarose gel electrophoresis. Lane M, 100 bp DNA ladder. *Gh2* to *gh100* represent PCR-amplified *spa* strains (180–600 bp).

**Figure 2 antibiotics-08-00202-f002:**
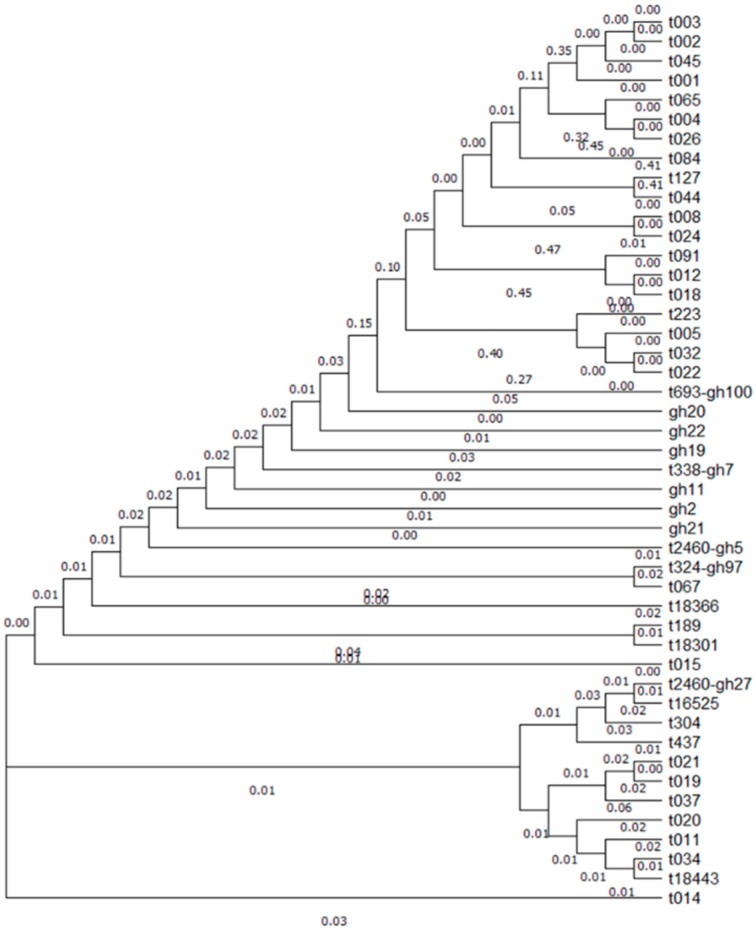
A split circle showing the similarity of the DNA sequence cluster. A total of 35 current strains of methicillin-resistant *Staphylococcus aureus* (MRSA) and 11 isolated strains from patient samples in Korea. The image provides an overview of the 46 total split circles and a detailed view of the region containing the 26 highly similar samples, which includes 15 existing and 11 new isolates (Multiple Sequence Alignment by CLUSTAL2.1, http://rest.genome.jp/link/. Kyoto University Bioinformatics Center, Kyoto, Japan).

**Figure 3 antibiotics-08-00202-f003:**
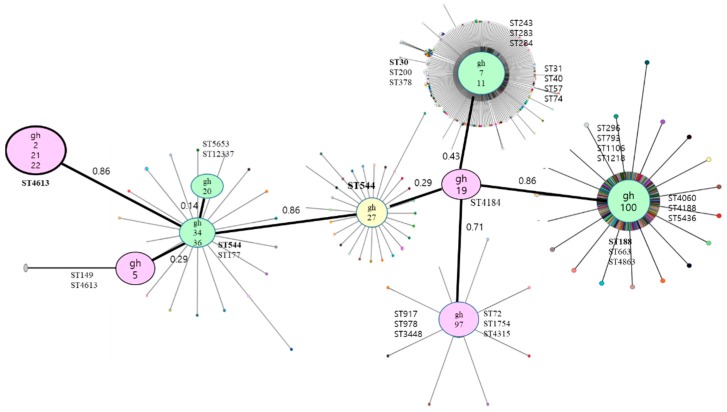
Output of an eBURST analysis of *S.*
*aureus* sequence types (ST) in the multi-locus sequence typing (MLST) database, listed as being MRSA of human origin. The group definition was set to “zero alleles in common” in order to allow visualization of all STs on a single diagram. The diameters of the solid spots are proportional to the occurrences of the STs in the MLST database (i.e., a large spot means that a large number of isolates with that ST have been entered into the database). Red is multi-drug resistance (MDR), green is no MDR.

**Table 1 antibiotics-08-00202-t001:** Phenotypic antibiotic resistance patterns and rates of antibiotic-resistant genes in *S. aureus* samples.

Samples	Meth	Pen	Kan	Erh	Gen	Tet	Strep	Van	Chlo	AML	TC	PRL	FEP	CTX	CAZ	IMI	ETP	MRP	ATM
gh2	0	0	0	0	0	10	8	15	22	20	14	0	0	0	0	7	0	10	0
gh22	0	0	0	0	0	11	8	14	21	14	7	0	0	0	0	0	0	7	0
gh11	16	0	0	20	0	23	8	14	25	25	25	25	25	25	25	25	25	25	0
gh19	0	0	0	0	0	12	10	14	25	12	0	0	0	0	0	0	0	0	0
gh20	15	0	12	20	12	20	8	12	23	24	25	25	25	25	25	25	25	25	0
gh21	0	8	12	0	10	8	8	13	20	18	13	0	0	0	0	0	0	9	0
gh34	18	9	17	20	15	23	10	13	22	25	25	25	25	25	25	25	25	25	0
gh36	0	0	17	23	15	25	10	15	25	15	18	12	22	18	15	14	18	20	0
gh5-t2460	0	0	11	0	11	10	0	15	22	14	10	10	0	0	0	0	0	8	0
gh7-t338	16	0	14	9	12	25	8	14	24	25	25	25	25	25	25	25	25	25	25
gh27-t2460	0	0	0	22	0	25	9	14	23	14	20	14	20	20	14	25	25	25	0
gh97-t324	0	10	8	27	18	28	15	19	29	13	8	9	15	14	0	0	0	14	0
gh100-t693	19	20	21	29	20	28	14	16	29	18	20	14	15	15	9	25	20	25	0

We tested the following 19 antibiotic discs (Liofilchem, Roseto degli Aburzzi, Italy). We measured the diameter (mm) of the inhibition zones and determined each isolate as resistant or susceptible to antimicrobial agents based on CLSI guidelines and criteria and Liofilchem quality control.

**Table 2 antibiotics-08-00202-t002:** Comparison of previously known Ridom types and novel *spa* types in *S. aureus* samples.

samples/ Ridom type	repeat sequence	repeat units	length	spa repeat sequence	spa repeat sequence
gh2	41	7	168	T1:M1:B1:B1:M1:E1:*	r26:r17:r34:r34:r17:r13:r314
gh22	31	7	168	T1:M1:B1:B1:M1:E1:K1	r26:r17:r34:r34:r17:r13:r16
gh11	41	8	192	U1:G2:M1:F1:*:B1:L1:B1	r07:r06:r17:r21:r81:r34:r22:r34
gh19	41	7	168	T1:M1:B1:B1:M1:E1:*	r26:r17:r34:r34:r17:r13:r314
gh20	44	6	144	U1:J1:G1:F1:*:*	r07:r23:r12:r21:r298:r254
gh21	47	10	240	T1:M1:B1:B1:*:*:*:*:*:*	r26:r17:r34:r34:r298:r377:r298:r298:r298:r314
gh5-t2460	25	10	240	T1:M1:B1:B1:M1:D1:M1:M1:M1:K1	r26:r17:r34:r34:r17:r20:r17:r17:r17:r16
gh7-t338	42	7	168	W1:F1:K1:A1:O1:M1:Q1	r15:r21:r16:r02:r25:r17:r24
gh27-t2460	35	10	240	T1:M1:B1:B1:M1:D1:M1:M1:M1:K1	r26:r17:r34:r34:r17:r20:r17:r17:r17:r16
gh97-t324	45	10	240	U1:J1:G1:G1:M1:D1:M1:G1:G1:M1	r07:r23:r12:r12:r17:r20:r17:r12:r12:r17
gh100-t693	48	1	24	U1	r07

* Ridom is spaserver.ridom.de, CLUSTAL 2.1 is multiple sequence alignment, gh is Gil Hospital, ND is not determined.

**Table 3 antibiotics-08-00202-t003:** Comparison of novel strains in S. aureus samples revealed through MLST.

*Samples*	*arcC*	*aroE*	*glpF*	*gmk*	*pta*	*tpi*	*yqiL*	*MLST*	*Clonal complex*
*gh2*	*1*	*4*	*1*	*4*	*559*	*495*	*10*	*ST4613*	*CC5*
*gh22*	*1*	*4*	*1*	*4*	*559*	*495*	*10*	*ST4613*	*CC5*
*gh11*	*1*	*4*	*1*	*ND*	*ND*	*ND*	*ND*	*ND*	*[CC1]*
*gh19*	*1*	*4*	*1*	*4*	*559*	*134*	*10*	*ND*	*[CC5]*
*gh20*	*3*	*1*	*1*	*8*	*322*	*495*	*295*	*ND*	*[CC1]*
*gh21*	*1*	*4*	*1*	*4*	*559*	*495*	*10*	*ST4613*	*CC5*
*gh34*	*1*	*4*	*1*	*8*	*4*	*497*	*76*	*ND*	*ND*
*gh36*	*177*	*4*	*1*	*8*	*4*	*368*	*76*	*ND*	*ND*
*gh5-t2460*	*1*	*4*	*1*	*4*	*559*	*41*	*10*	*ND*	*[CC45]*
*gh7-t338*	*2*	*2*	*95*	*185*	*6*	*201*	*500*	*ND*	*[CC1]*
*gh27-t2460*	*1*	*4*	*1*	*185*	*4*	*497*	*76*	*ND*	*[CC1]*
*gh97-t324*	*1*	*4*	*1*	*8*	*4*	*497*	*3*	*ND*	*[CC8]*
*gh100-t693*	*3*	*1*	*1*	*8*	*1*	*134*	*295*	*ND*	*[CC1]*

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
