# Peer review of "Novel spa and Multi-Locus Sequence Types (MLST) of Staphylococcus Aureus Samples Isolated from Clinical Specimens in Korean"

_antibiotics, 2019, doi:10.3390/antibiotics8040202_

Round 1

Reviewer 1 Report

Authors have done a lot of work, and their research is good. However, somehow their presentation of results lack cohercoherence. It is mainly due to poor usage of English. Sentrnce composition is very poor, making it difficult to understand. As I mentioned in my previous comments on the manuscript (original version).    My suggestion will  be to ask authors do two things, 1. Ask their peers to read this manuscript and give comments. Peer reader could preferably be a person who had published articles and who understand bacteriology.  2. With inputs from their peer group, manuscript should be sent for English-language editing with a native speaker. 

Author Response

Manuscript ID#.: antibiotics Manuscript ID: antibiotics-617577 - Major Revisions 

Title: “Novel spa and mlst Types of Staphylococcus aureus Samples Isolated from Clinical Specimens in Korean

Title is insert words: Novel spa and Multi-Locus Sequence Types (MLST) of Staphylococcus aureus Samples Isolated from Clinical Specimens in Korean

Comments by Reviewer #1

Open Review

Comment #1.

Comments and Suggestions for Authors

Authors have done a lot of work, and their research is good. However, somehow their presentation of results lack cohercoherence. It is mainly due to poor usage of English. Sentrnce composition is very poor, making it difficult to understand. As I mentioned in my previous comments on the manuscript (original version).    My suggestion will  be to ask authors do two things, 1. Ask their peers to read this manuscript and give comments. Peer reader could preferably be a person who had published articles and who understand bacteriology.  2. With inputs from their peer group, manuscript should be sent for English-language editing with a native speaker. 

Submission Date

30 September 2019

Date of this review

09 Oct 2019 06:22:49

Response #1. Thank for your comment. We have again proofreading this paper with corrective native English Editing.

Response #2. Thank for your careful consideration of the manuscript.

We have corrected abstract, Introduction, results, discussion, materials and methods.

We have changed final references, tables and figures.

Reviewer 2 Report

lines 23-24: "This gh22 isolate was identified in 15 antimicrobial susceptibility tests" - not understandable. English needs revision in the end of the abstract.

73 - authors mention "clinical sources", but in methods (line 252) is about blood cultures only  - this has to be clarified.

Most of the phrases are incomplete or English need revision - complete revision is needed.

269 - please delete "extended spectrum"

S. aureus should be written with italics all over the text.

Methods: antibiotic susceptibility was performed on lysogeny agar - according to CLSI, has to be performed on Muller-Hinton agar incubated at 35°C.

266-274 - CLSI does not have standards for many of the enumerated antibiotics for S. aureus - so how were the diameter breakpoints interpreted?

272 - delete <=10-13

Author Response

Response to the Comments

Manuscript ID#.: antibiotics Manuscript ID: antibiotics-617577 - Major Revisions 

Title: “Novel spa and mlst Types of Staphylococcus aureus Samples Isolated from Clinical Specimens in Korean

Title is insert words: Novel spa and Multi-Locus Sequence Types (MLST) of Staphylococcus aureus Samples Isolated from Clinical Specimens in Korean

Comments by Reviewer #2

Comment #1.  

Comments and Suggestions for Authors

lines 23-24: "This gh22 isolate was identified in 15 antimicrobial susceptibility tests" - not understandable. English needs revision in the end of the abstract.

Response #1. Thank you for your pointing out.

We have changed this words.  This gh22 isolate was identified in antimicrobila susceptibility tests of 15 kinds of antibiotics.

Comment #2.  

73 - authors mention "clinical sources", but in methods (line 252) is about blood cultures only  - this has to be clarified.

Response #2. Thank for your careful consideration of the manuscript.

We have obtained samples with human source (blood, sputum, urine, etc)

Comment #3.  

Most of the phrases are incomplete or English need revision - complete revision is needed.

269 - please delete "extended spectrum"

Response #3. Thank for your careful consideration of the manuscript.

We have changed amoxicillin (25 μg) 

Comment #4.  

aureus should be written with italics all over the text.

Response #4. Thank for your careful consideration of the manuscript.

We have changed S.aureus.

Comment #5.  

Methods: antibiotic susceptibility was performed on lysogeny agar - according to CLSI, has to be performed on Muller-Hinton agar incubated at 35°C.

Response #5. Thank for your careful consideration of the manuscript.

We have chaged : Muller-Hinton agar, and incubated in the presence of antibiotic discs at 35 °C for 18 hours.

Comment #6.  

266-274 - CLSI does not have standards for many of the enumerated antibiotics for S. aureus - so how were the diameter breakpoints interpreted?

Response #6. Thank for your careful consideration of the manuscript.

266-274 - CLSI does not have standards for many of the enumerated antibiotics for S. aureus - so how were the diameter breakpoints interpreted?

 We have changed the words and we recommed with Liofilchem company and CLSI guidelines. 

Comment #7.  

272 - delete <=10-13

 Response #7. Thank for your careful consideration of the manuscript.

272 - delete <=10-13

 We have deleted word.  

Round 2

Reviewer 2 Report

Most of the reported issues were addressed. Though, minor corrections are still needed (see line 187 - only blood culture is mentioned), English quality - mostly grammar - should be further improved.

Author Response

Response to the Comments 2

Manuscript ID#.: antibiotics-617577

Title: Novel spa and mlst Types of Staphylococcus aureus Samples Isolated from
Clinical Specimens in Korean

 Comments by Reviewer #1

 Comment #1. Comments and Suggestions for Authors

Most of the reported issues were addressed. Though, minor corrections are still needed (see line 187 - only blood culture is mentioned), English quality - mostly grammar - should be further improved.

Submission Date 30 September 2019

Date of this review 19 Oct 2019 09:27:50

Response #1. The authors very much appreciate your comment on repeat words. We have changed the sentence.

Thank for your comment, and for the future, we will improve English writing and editing.

: Blood culture - blood agar plates (Shin Yang Chemical Co., Ltd.’s media., Seoul, Korea)

This manuscript is a resubmission of an earlier submission. The following is a list of the peer review reports and author responses from that submission.

Round 1

Reviewer 1 Report

This work is an interesting study that identifies a new strain of S. aureus, and the scientific content is sound, however numerous issues need to be addressed before it is suitable for publication.

First, the intro does not seem to adequately contextualize the importance of this work.  We are all aware that MRSA is a threat to human health, but what specifically is gained from this study?  Overall the motivations appeared unclear.

Figures 2 and 3, especially Figure 2 was entirely unintelligible and unreadable.  The font needs to be changed, or some information needs to be highlighted.  It was impossible to tell what the authors were trying to convey. Figure 2 especially must be fixed before this paper is accepted for publication.

The clarity of Table 1 would be greatly improved by color coding via a heat map.

Reviewer 2 Report

Manuscript submitted by Mun et al, entitled “Novel Spa Type of Staphylococcus aureus Strain Isolated from Clinical Specimens in Korean” describe prevalence of various S. aureus strains in Korea. Authors collected clinical specimen from one hospital and tested them against a panel of 19 antimicrobial drugs for their resistance. The drug resistant specimens were further subjected to PCR and sequencing, which led to identification of a various new S. aureus protein A (Spa) types. Thus, adding in our understanding and prevalence of multiple drug resistant bacteria in different regions of the globe.

While study is interesting and add details in MRSA pathogenesis and distribution, the manuscript is very poorly prepared. Presentation of results is not in order, and at several places this reviewer is not clear of the intended message conveyed. Therefore, manuscript should be thoroughly proof-read, and several sections be re-written. My comments, detailed below, may be helpful in adding value to the manuscript.

1.    Title of the manuscript should have “Korea” rather than “Korean”. It should highlight use of MLST (mutli-locus sequence typing), which is the key-point in this study and authors failed to recognize this.

2.    Authors need to be consistent in using some terminology, e.g. Spa vs spa. Also they should be thoughtful while using “clone” “isolates” “strains” “types”. These terms had been used to describe same thing, thus confusing what is what.

3.    Reference no. 1 is quite old.

4.    Introduction is insufficient in addressing the need for this study. Authors did not establish how and why identification of new types of Spa is helpful in containment of the disease. Terms like antibiotic sensitivity testing should be established while describing MRSA.

5.    The experimental design of the study is not clear. Authors studied 109 strains of S. aureus (line 62), but in methods they describe analysis of 109 isolates (line 172). Authors need to describe in chronological order how many clinical specimens were used, of them how and how many isolates or strains were identified. Of all the clinical specimens, how many are from same patient or from different patient. What method was used to identify “isolate” or “strain”. Following this identification what was the next step: antibiotic drug resistance test, PCR, sequencing? A flow-chart describing experimental order will be helpful.

6.    In line 63, authors selected “11 specific samples”. What does it mean? 11 “strain” or “isolate” or clinical specimen? What was the criterion for their selection?

7.    Authors need to explain why the six new Spa types were considered new? Only based on definition by Ridom Spa Server? Why those new “types” were not confirmed with pulsed-field gel electrophoresis, which is considered as gold standard (line 46)?

8.    Since identification of six new Spa “types” is novelty of the study, authors should provide detailed methods including primer sequences used and other relevant information.

9.    Paragraph from Line 83 to 95 is not clear at all. The repeat sequences had been shown in table 3, repeating them in results and again in discussion sections is not helpful.

10. Presentation of results should be chronological. The experimental order flow char, as noted in point 5 above, should be followed to write results section. In present format it is very confusing.

11. Table 1, need details of abbreviations used to while naming various antimicrobial agents. Authors should be careful while using term antibiotic or antimicrobial.

12. Table 1, statement like “CLSI guidelines and criteria” are not enough. Describe in brief how experiment was conducted and interpreted.

13. Table 1, 2 and Figure 1, all need to include data from known Spa types e.g. t2460 etc.

14. Table 2, blank spaces should be replaced by “ND” (nothing detected). Also, figure legend should have brief methodology used.

15. Table 3, what does “*” mean? “ridom” should be “Ridom”. Authors need to explain what does repeat sequence mean, and why there are two types of repeat sequence. What do they represent? In present format it is abstract information.

16. Fig1, describe primer pairs used, with expected product length. Gel image 2 has some of genes not described in the manuscript. E.g. gh63, gh97.

17. Fig 2, “gh-2” vs “gh2”. Be consistent in terminology usage. Fig 2 is too small to get details. Should have include sequences of newly identified types.

18. Fig 3, line 121, should emphasize 11 new isolates which are identified in this study.

19. Authors need to be certain how many types had they identified, “five” (line 136, 162), “six” (line 75), or “11” (line 121, 129?

20. Authors need to describe what was unique in their method which led them to identify these new types? Why they were not identified previously?

21. Line 141, authors discuss “SCCmec type” for the first time in the manuscript. What is the rationale?

22. Line 152-157, repeat of results section. Very poor.

23. Discussion section lacks any meaningful discussion and does not tell about newly identified types.

24. Line 173, what is previous report?

25. Line 176, “Most of the chemicals…”, be precise. What chemicals?

26. Line 179, how S. aureus “strains” were confirmed? Describe.

27. Why authors selected only 19 antimicrobials? Describe.

28. Line 191, “we developed…” it is an ATCC isolate? Describe.

29. Line 202, show a list of primers used in identification of various genes. List should include PCR primers, as well as sequencing primers.

30. Line 208-212, is unnecessary. Agarose gel electrophoresis is an established technique.

31. Line 228, funding should be included in “Funding” section.

32. Appendix A, what is the purpose. It has not been referenced in the manuscript.

33. Reference numbering is repeated. Similarly, Table number is repeated.